# Two Dimensional β-InSe with Layer-Dependent Properties: Band Alignment, Work Function and Optical Properties

**DOI:** 10.3390/nano9010082

**Published:** 2019-01-09

**Authors:** David K. Sang, Huide Wang, Meng Qiu, Rui Cao, Zhinan Guo, Jinlai Zhao, Yu Li, Quanlan Xiao, Dianyuan Fan, Han Zhang

**Affiliations:** 1Shenzhen Key Laboratory of Two Dimensional Materials and Devices, Shenzhen Engineering Laboratory of Phosphorene and Optoelectronics, International Collaborative Laboratory of 2D Materials for Optoelectronics Science and Technology, College of Optoelectronic Engineering, Shenzhen University, Shenzhen 518060, China; dks@szu.edu.cn (D.K.S.); wanghuide@szu.edu.cn (H.W.); qiumeng@szu.edu.cn (M.Q.); caorui@szu.edu.cn (R.C.); zhaojl@szu.edu.cn (J.Z.); xiaoql@szu.edu.cn (Q.X.); fandy@cae.cn (D.F.); 2College of Materials Science and Engineering, Shenzhen University, Shenzhen Key Laboratory of Special Functional Materials, Shenzhen 518060, China

**Keywords:** Layer-dependent, Indium Selenide, density functional theory, work function, optical properties

## Abstract

Density functional theory calculations of the layer (L)-dependent electronic band structure, work function and optical properties of β-InSe have been reported. Owing to the quantum size effects (QSEs) in β-InSe, the band structures exhibit direct-to-indirect transitions from bulk β-InSe to few-layer β-InSe. The work functions decrease monotonically from 5.22 eV (1 L) to 5.0 eV (6 L) and then remain constant at 4.99 eV for 7 L and 8 L and drop down to 4.77 eV (bulk β-InSe). For optical properties, the imaginary part of the dielectric function has a strong dependence on the thickness variation. Layer control in two-dimensional layered materials provides an effective strategy to modulate the layer-dependent properties which have potential applications in the next-generation high performance electronic and optoelectronic devices.

## 1. Introduction

Two dimensional layered materials (2DLM) are kind of materials with layered structure which can be obtained by exfoliation of layered bulk materials due to the weak interlayer binding energies [1]. Besides graphene, hexagonal boron nitride (h-BN) [2], transitional metal dichalcogenides (TMDs) [3,4] and black phosphorus (BP) [5,6], are typical 2DLMs. Because of their atomic-level thickness, quantum confinement gives 2DLMs unusual electronic properties, optical properties and thermal conductivity, in contrast to their bulk counterparts. These novel 2DLMs have opened a new twist in understanding the science and technology in nanomaterials.

Although, not each kind of 2DLMs possesses layer dependent physical properties, thickness is one of the most important parameters and should be perfectly controlled when 2DLMs are used in either fabrication of electronic or optical devices. Monolayer and few-layer TMDs such as MoS_2_, WS_2_, MoSe_2_ and WSe_2_ usually exhibit a thickness-induced indirect-to-direct band gap [7,8]. The excellent optical properties at monolayer level have gained a lot of attraction in terms of applications in lasers and 2D-light-emitting diodes (LEDs) [9,10]. Monolayer WSe_2_ exhibits strong optical absorption in the visible range and good light-to-electricity conversion coefficient of approximately 0.5% [11]. The MoS_2_-graphene heterojunction with an optimized thickness ratio manifest an enhanced energy conversion coefficient of up to 1% [12], giving rise to potential applications in solar energy conversion devices [13]. Moreover, MoS_2_ and WS_2_ with different thicknesses have been studied for their potential use in photodetectors [14], valleytronic [15] and spintronics devices [16]. In this case, accurately predicting the physical properties of 2DLMs with exact layer number and obtaining the rule of the layer dependent properties is very important for directing the design and fabrication of the electronic and optoelectronic devices.

Recently, InSe was added into the family of layered transitional metal monochalcogenide. InSe exhibits high photoresponsivity, excellent electrical properties and nonlinear effect and with these extraordinary properties, InSe has captured a lot of attention in the last few years [17,18,19,20]. The greatest milestone is the achievement of liquid-phase exfoliated InSe flakes [21,22,23], which has provided a platform to investigate exciton physics phenomena and engineered practical devices for novel applications in optoelectronic and nanoelectronic field. Recently, Synthesis, electronic properties, ambient stability and applications of InSe were reported [24]. The InSe flakes are not susceptible to oxidation [25,26,27,28,29], even if the Se vacancies induces chemical reactivity towards water [25]. InSe crystals can exist in three polytypes denoted as β, γ and ε phases. γ-InSe is the mostly studied polytype with ABCABC stacking arrangement. Monolayer and few-layer γ-InSe have been proved to possess high electron mobility in the order of 10^3^ cm^2^ V^−1^ S^−1^ [30,31], excellent metal contact and moderate band gap range [32], which offer the opportunity for presenting tunable nanodevices [33,34,35,36]. Sanchez-Royo et al. previously reported a work on γ-InSe, which shares the same composite as β-InSe but stacking in different arrangement [37] and found that there was a huge increase of electronic band gap by more than 1 eV for a single layer and is in agreement with this work, despite the fact that they were investigating the band structures of γ-InSe using DFT approach as implemented in SIESTA code [38]. However, the intrinsic instability of γ-InSe hinders its practical application either in electronics or optoelectronics [39]. The ε–InSe has ACAC layer arrangement and exhibits indirect band structure with a band gap value of 1.4 eV with high photoresponsivity of 34.7 mA/W in few-layer structure [35]. The β-InSe is of great significant since it has been exfoliated into individual layer with hexagonal structure [40], with different electronic and optical properties compared to the bulk β-InSe and it is the most stable phase of InSe due to the ABAB crystal stacking mode [41]. Monolayer and few-layer β-InSe possesses moderate band gap of 2.4 eV and 1.4 eV, respectively [35], which can be an optimal candidate for use in the broadband optoelectronic devices. Moreover, appreciable shift of valence band maximum (VBM) upon thickness variation could be very important for optimizing the band gap to improve electrons and holes mobility. Although, several studies on β-InSe have been carried out to comprehend the tunable performances of the electronic and optoelectronic devices fabricated from it with different layer number and it is important to understand the intrinsic properties which are layer-dependent, that is, band alignment, work function and optical properties.

In this study, the electronic band structures, work function and optical properties of β-InSe monolayer, few-layer and bulk β-InSe have been investigated by performing first principle calculations as implemented in VASP 5.4 [42]. Electronic band structure, work function and optical properties of β-InSe have been demonstrated to be layer thickness-dependent and this provides a wide range of tunability of band gap and corresponding electronic properties, work function and optical properties. The results from this study would provide guideline to experimentalists in obtaining optimal parameters for the design of nanoelectronic and optoelectronic β-InSe-material-based devices.

## 2. Methods

First-principles calculations based on the density functional theory (DFT) in generalized gradient approximation (GGA) [43], has been performed with the Perdew-Burke-Ernzerhof (PBE) functional [44] for electron exchange-correlation potentials as implemented in the Vienna ab initio Simulation Package (VASP 5.4). The electron-ion interaction was described by employing the projector augmented wave (PAW) method [45] and the cutoff energy for the plane-wave basis was set to 500 eV. To account for the interlayers interaction in few-layer InSe (L > 1), we used van der Waals (vdW) correction proposed by Grimme (DFT-D2) [46]. The Brillouin zone with a Monkhorst-Package scheme 10 × 10 × 1 for *k*-point grid for sampling during structure optimizations and 16 × 16 × 1 in single point calculations was used. The structures were fully optimized via the conjugated gradient algorithm until the equilibrium configuration of atoms was less than 0.01 eVA^−1^. The convergence criteria energy of electronic in SCF cycles was set to be 10^−5^ eV. In order to mimic the two-dimensional system and to avoid/or make it negligible interaction between repeated unit cells, a vertical separation vacuum space of at least 16 Å was created in the unit cell in the *z*-direction perpendicular to the 2D surface during all calculations. The Phonon property of monolayer β-InSe was calculated using the Full Brillouin zone method implemented in Phonopy [47]. A 4 × 4 supercell was constructed to calculate the atomic forces employing VASP 5.4, with electronic convergence set to 10^−8^ eV using the normal (block Davidson algorithm).

To confirm the thermal stability of 2D-InSe monolayer, ab initio molecular dynamics (AIMD) simulations within the framework of NVT ensemble (constant number of particles, volume and temperature) [48] was performed. To observe changes in 2D-InSe monolayer structure at the atomic level in the present equilibrium state, a cell with same length, a = b in the x and y direction was considered, respectively. The monolayer was calculated with 4 × 4 × 1 supercell, with 64 atoms in total. Sampling configuration space was carried out at temperature between 300 K and 1000 K. The valence electrons from 4d^10^5s^2^5p^1^ for In atom and 3d^10^4s^2^4p^4^ for Se atom orbitals are included.

## 3. Results and Discussion

### 3.1. Crystal Structure

Bulk InSe can exist in three polytypes denoted as β, γ and ε phases, which show ABAB, ABCABC and ACAC stacking order, respectively. In this work, we focused on the most energetically favorable β-InSe phase, for which the 2DLM crystallizes to form hexagonal structure stacked in AB order, as is shown in Figure 1. The unit cell of β-InSe comprises a base-centered hexagonal lattice classified in the space group P6_3_/mmc, $D_{6h}^{4},$ number 194. From the experimental work [49], lattice constants are *a* = 4.05 Å, *b* = 4.05 Å, *c* = 16.93 Å.

### 3.2. Electronic Band Structures of Monolayer and Few-Layer of β-InSe

The study on electronic properties of β-InSe is very significant since it gives insightful description of the system. Owing to the manifestations of the quantum size effects (QSEs) in β-InSe, the nature of the band structure exhibits indirect transitions in monolayer and few-layer. In the scope of this work, E_F_ = 0, is set to be the center of the energy gap. The band gap values of the β-InSe monolayer (1 L), few-layer (2 L to 9 L) and bulk β-InSe within the GGA were tabulated as shown in Appendix A. Calculations in this work showed that, bulk β-InSe, monolayer and few-layer are semiconductors due to the present of band gap in their electronic band structures as shown in Appendix A. Due to high computation demand from HSE06 functional, only 1 L, 3 L, 5 L and bulk were selected for this study and because GGA-PBE underestimate the band energy value, HSE06 was opted as the best choice because of its accuracy. The calculated band gap values based on HSE06 pseudo-potential for 1 L, 3 L, 5 L and bulk are 2.84 eV, 1.98 eV, 1.84 eV and 1.39 eV, respectively. It is observed that, bulk β-InSe possess direct allowed transition and its band gap energy is comparable with the reported experimental results (1.2~1.30 eV) obtained by use of photoemission electron spectroscopy (ARPES) [50,51] and the β-InSe monolayer (1 L) and few-layer (3 L and 5 L) exhibit robust indirect band gap character as shown in Figure 2. The band gap energy of 1 L (2.84 eV) is comparable with the result from tight-binding model with scissor correction [52]. Magorrian et al. demonstrate that inclusion of spin orbital coupling (SOC) in InSe system has little effect on energy band gap and in this work, it was cross check with monolayer β-InSe and established to be in agreement, as shown in Appendix A, where the difference is 0.02 eV.

The minimum point of conduction band (CBM) appears at the Г point and is parabolic in nature and the maximum of the first valence band (VBM) appears at a point along the Г-F direction. Another key character in these band structures is the two peaks at the edge of the valence band maximum (VBM). The two peaks (the so-called sombrero-shape dispersion) at edge of valence band enhances the probability of electron transfer between the valence band and conduction band in monolayer and few-layer, more so for optical conductivity as compare to bulk β-InSe and this fascinating optical response is also highly exhibited by GaS [53].

Moreover, a significant point to note in β-InSe band structures is that, there exist more than one valence band maxima and conduction band minima in almost the same momentum vector, thus electronic allowed transition can take place in these extrema via optical absorption. On the other hand, the band dispersions observed in monolayer, few-layer and bulk are similar and are attributed to uniform crystallinity but the dispersions intensity are quite not the same and this is link to thickness dependency. The symmetric nature of the band dispersion along the high symmetry points of F and Г in all band structures shows symptomatic isotropic behavior of electronic properties of monolayer and few-layer of β-InSe like graphene, MoS_2_ and h-BN layers. The band gap dependence on layer thickness offers tunability of the band gap as well as the corresponding associated electronic properties which is crucial for smart electronic and optoelectronic devices. In order to understand the contribution of different orbitals to the electronic states, calculations of the total DOS and the partial DOS for β-InSe monolayer, few-layer and bulk-β-InSe was performed. The Fermi level in this work is set to E_F_ = 0 eV and is shown by green horizontal dash lines. As shown in Figure 3, the states at the bottom of the VBM have contribution from both p states of In and Se atoms, with little contribution from s states of In and Se atoms. The states forming CBM is mainly derived from the s orbitals of In and p orbitals of Se atoms.

Despite the fact that the valence band have p orbitals from both In and Se atoms, all these orbitals hybridized to form an orbital which is close to the Fermi level energetically. The partial density of states (PDOS) of In and Se atoms shows further evidence in the band hybridization in 1 L, 3 L, 5 L and bulk β-InSe as shown in Figure 3. This hybridization (sp^3^) depicts a strong covalent interaction which is a chemical bonding. The trend in the DOS calculation shows that 2D-DOS remained almost constant, giving insights that the electronic states of β-InSe monolayer (1 L), few-layer (2 L to 9 L) have weak dependence on the layer thickness as shown in Appendix A. The conduction bands of few-layer β-InSe depict finite and almost constant 2D-DOS within a small range of energies due to quantum confinement effects, unlike bulk β-InSe which shows a slight change in DOS within the conduction band region. The valence band in the DOS exhibit a slight shift towards the Fermi level as the layer thickness increase and this shows the effect of layer modulation in β-InSe.

Since layer control has been used to regulate the electronic properties of semiconductors, also in this study, it gives insights on how the control thickness influences the work function. Work function has been described as energy needed to remove one electron from the system. Therefore, it can be calculated by subtracting the Fermi energy from the electrostatic potential in the middle of the vacuum. From the Figure 4c, it shows that, layer thickness has a strong effect on work functions of β-InSe and hence can be a powerful parameter in controlling the work functions. When the thickness was reduced from bulk to monolayer, the work function of the β-InSe increases monotonically from 4.77 eV to 5.22 eV and this is due to the quantum size effect existing in this atomically thin material (β-InSe monolayer) which make the binding energy of exciton very strong than in bulk β-InSe. The change in work function values as the layer thickness is varied; gives insights on the proper selection of appropriate contacting material with β-InSe and this will offer opportunity for tuning the Schottky barrier to optimal, which in turn improves the carrier mobility.

The β-InSe monolayer and few-layer showed that, the band gap values increase with decrease in the number of layers following a power law as depicted in Figure 4a, showing the variation in the electronic band gap as a function of the layer thickness of β-InSe based on various functionals. The alignments of conduction band minimum (CBM) and valence band maximum (VBM) wavefunctions with respect to the vacuum level are shown in Figure 4b for monolayer (1 L), few-layer β-InSe (3 L and 5 L) and bulk β-InSe, based on hybrid functional calculations. The positions of CBM and VBM for 1 L, 3 L, 5 L and bulk β-InSe are −3.5 eV, −4.0 eV, −4.1 eV, −4.3 eV and −5.5 eV, −3.2 eV, −5.1 eV, −5.0 eV, respectively. The general trend shows the VBM energies shift upwards as the numbers of layers increases and CBM shift downwards. The tuning of the VBM and CBM energies in β-InSe via layer control offers a practical option to optimize the Schottky barrier height, hence improve electron injection efficiency, which leads to more efficient electron mobility across the contact.

To establish the dynamic stability of monolayer InSe, phonon dispersion shown in Appendix A was calculated within the framework of density functional perturbation theory. The phonon dispersion of monolayer β-InSe showed no imaginary lines in the first Brillouin zone, which is a confirmation that β-InSe is dynamically stable, suggesting the likelihood to be obtained as isolated stable layer.

The performed Molecular dynamics based on direct calculation of the coordinates and velocities of large ensemble of atoms as they evolve over a period of time, gives insightful information on changes in a material’s structure at atomic level in the present state of balance. In Appendix A, confirmed the thermal stability of β-InSe monolayer. Upon the sample was heated at 300 K temperature for 1 ps with a time step of 1 fs, it exhibits structural stability (no structural disorder). Also, it showed that β-InSe can withstand high temperature (1000 K) with no phase change, hence thermodynamically stable. In principal, this is a confirmatory of In-In and In-Se bonds in the β-InSe monolayer that, they are very strong and remained unbroken even when subjected to high temperature and this demonstrate that β-InSe can exist at 300 K and 1000 K. The high thermal stability of β-InSe can be a good material for electronic devices which operates at wide range of temperatures.

### 3.3. Optical Properties

To advance the materials, in order to gain and retain the recognition in industrial applications, particularly in the optoelectronic field, the optical properties of materials need continuous improvement. The polarizability of the material and polarization directions are the main factors which influence the optical response of the system under study. The polarization of the electric field of incident light (photon) is important and all the optical functions and spectra are calculated along the directions of x and z and are presented relative to hexagonal axis as extraordinary E || c and ordinary E ⊥ c waves, respectively.

The optical properties of β-InSe were studied via dielectric function. The dielectric function is the summation of real and imaginary parts. The function is as follow:ε (ω) = ε_1_ (ω) + ίε_2_ (ω)(1)

This function classifies the material response upon interacting with the electromagnetic spectrum. The real part (ε_1_ (ω)) was calculated using the Kramers-Krӧningrelation [54]. This relationship is derived from the framework of random phase approximation (RPA) method [55].
(2)$$\varepsilon_{1}\;\left(\omega \right)=1+\frac{2}{\pi }\;p\;\int_{0}^{\infty }\frac{ώ\varepsilon_{2}\left(ώ\right)}{{ώ}^{2}-\omega^{2}}\;\partial ώ$$
where *p* is the fundamental value (principal value) of this function and other notations are the same as the one in dielectric function.

In Figure 5a, the spectra of real part, ε_1_ (ω), in monolayer (1 L) merged at low energy slightly above 0 eV and the spectra in few-layer (3 L and 5 L) set apart at low energy slightly above 0 eV and the separation gradually increase. Compared to bulk β-InSe, the real part is slightly bigger than in few-layer β-InSe. This phenomenon is attributed to the reduced dimensionality and reduced dielectric screening which initiate dramatic change in band gap near the Fermi-level. The energy shift in monolayer and few-layer can provide an opportunity for polarization window be adjusted via layer control, thus β-InSe can be optical linear polarizer. The distinctive optical peaks in spectra remained unchanged as shown in Appendix A. The real part of the dielectric function is negative at energies between 4–6.5 eV in z-direction (out-of-plane) and 4.5–6.5 eV in x-direction (in-plane), respectively. Only monolayer does not show negative in z direction and this is attributed to the mirror symmetry in the monolayer crystal. The optical function that is, real part (ε_1_ (ω)) shows strong dependence on the direction of the electromagnetic wave as shown in Figure 5.

On the other hand, the inter-band momentum allowed transition of electrons is the most important key parameter of the optical spectra characterizer in the semiconductor materials and can be identified in the imaginary part of the dielectric function. To calculate the imaginary part of the dielectric function of β-InSe and bulk β-InSe, need to integrate all possible allowed transitions that is, only from valence to conduction band and to factor in the polarization of the electric field of incident light (photon) which is very significant and therefore, all the optical functions and spectra are calculated along the direction of x and z polarization and are presented relative to hexagonal axis as extraordinary E || c and ordinary E ⊥ c waves, respectively. Therefore, the imaginary part of the dielectric function is given by the equation:(3)$$ί\varepsilon_{2}\;\left(\omega \right)=\frac{4\pi e^{2}}{m^{2}\omega^{2}}\sum_{c,v}\int_{B}^{Z}d^{3}k\;{\left|\langle v_{k}|p_{2}|c_{k}\rangle \right|}^{2}\;x\;\delta \;\left(E_{ck}-\;E_{vk}-\hbar \omega \right),$$
where the term 〈*v_k_***|***p**_2_*****|***c**_k_***〉 consist of the occupied and unoccupied states of electrons in the valence and conduction bands. The *e* and *m* are the electron charge and mass, respectively. And ℏω is the energy of the incident photon. Since $\varepsilon_{xx}=\varepsilon_{yy}\neq \;\varepsilon_{zz}$ thus imaginary function $ί\varepsilon_{2\;}\left(\omega \right)$, can be substituted with $\varepsilon_{xx}$ and $\varepsilon_{zz}$ and the fundamental value $\;p_{2}$, take the form $p_{xx}$ and *p_zz_* The imaginary parts of dielectric function of β-InSe monolayer, few-layer and bulk β-InSe against photon energy (eV) are shown in Figure 6.

In Figure 6, the peaks are directly connected to different inter-band transitions in the Brillouin zone and the induced evolution mechanism of the band near the Fermi energy level. In Figure 6, there is a shift in energy from high to low as witness in monolayer (3.3 eV) and bulk β-InSe (3.0 eV), in E ⊥ c polarization, respectively. This demonstrates that, the control of layer thickness tends to change the frequencies of the absorbed photons and shows that a tunable optical response can be achieved in β-InSe by decreasing layer thickness down to monolayer. The shift into lower energy in β-InSe monolayer and few-layer is due to reduced perpendicular quantum confinement as a result of weak interlayer interaction and this yields weak excitons. Imaginary function (ίε_2_ (ω)) of β-InSe revealed a strong dependence on the direction of the electromagnetic wave as shown in Figure 6.

Other than qualitatively, optical properties can be used quantitatively. The absorptance can increase to a larger value depending on the photon energies range. In this work’s calculations, the photon energies range substantially increase in β-InSe monolayer to bulk β-InSe as seen in the peaks intensities and this subsequently increase the absorptance value. Therefore, layer control can be utilized in tuning the absorptance values. In the transparency region of β-InSe, the refractive indices of monolayer (1 L), few-layer (5 L) and bulk β-InSe are found to be 2.35, 3.0 and 3.75 in E ⊥ c and 2.3, 2.55 and 3.27 in E || c, respectively, as shown in the Appendix A and the calculated birefringence (Δn = *n*_e–_*n*_o_) are −0.05 for monolayer (1 L), −0.45 for few-layer (5 L) and −0.48 for bulk β-InSe and all values are negative, indicating that β-InSe is a negative single-axis crystal. The estimated transparency range in the E ⊥ c polarization direction has been established to be 0.30633 ± 0.04103 and 0.43 μm thickness is required to absorb fully monochromatic light. In the E || c polarization direction, the transparency has been estimated to be 0.32824 ± 0.01465 and the 0.38 μm thickness is required to absorbed fully monochromatic light.

In this work, the peak intensities of 1 L, 3 L, 5 L and bulk β-InSe that is, heights of fundamental peaks of (ίε_2_ (ω)) in E ⊥ c polarization have been predicted to be 4.8 (3.3 eV), 7.4 (3.25 eV), 8.5 (3.2 eV) and 14 (3.0 eV) respectively; where the position of the maxima over the photon energy scale is shown in parentheses. As shown in Appendix A, the general trends in peak intensities increase with increase in layer thickness. It reaches maximum in the bulk β-InSe and this is pronounced due to the direct exciton and direct transition. The shape of the peaks in both polarization directions remain constant, demonstrating that, variation in layer thickness do not affect the shape but the photon energies positions and intensities are layer dependent.

Indeed, due to layer control, the band gap increases and decreases depending on the number of layers and this is influenced by the quantum confinement size effects, resulting in the change of orbital overlapping. Thus, the change in the band gap definitely changes the photo-absorption energy that in turn leads to emission of electrons. Therefore, the variations in band gap leads to change in the threshold of imaginary and real part of the dielectric function.

## 4. Conclusions

We performed a first-principles study on layer dependent electronic band structures, work functions and optical properties of β-InSe, which give advantage in fine tuning the thickness-dependent behavior for possible utilization in the next-generation high-performance electronic and optoelectronic devices. The calculations based on HSE06 functional for 1 L, 3 L, 5 L and bulk has been found to be 2.84 eV, 1.98 eV, 1.84 eV and 1.39 eV, respectively. Bulk β-InSe depicts direct allowed transition and the β-InSe monolayer (1 L) and few-layer (3 L and 5 L) exhibits robust indirect band gaps. Work functions of the β-InSe increases monotonically from 4.77 eV (bulk) to 5.22 eV (monolayer) and understanding the strain effect induced by layer thickness control on work functions of β-InSe is of great important since it offers the achievable route to optimize the Schottky barrier height for future design and developing electronic devices of β-InSe-nanoscale-based devices with superior functionalities. Moreover, optical properties can be efficiently tuned by varying the number of layers as seen in a shift of energy from high (monolayer (3.3 eV) to low (bulk β-InSe (3.0 eV), in E ⊥ c polarization), which have appeared as exciting technique to tune the optical response in β-InSe. We hope, this study will stimulate further experimental and theoretical studies on this promising β-InSe material and can be extended to other 2D-materials.

## Figures and Tables

**Figure 1 nanomaterials-09-00082-f001:**
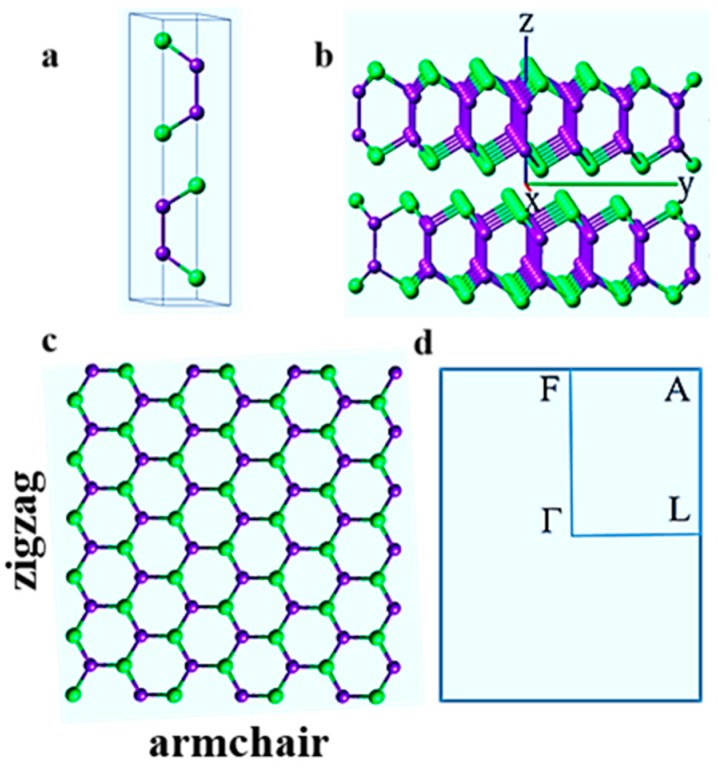
(**a**) Side view of the AB stacking arrangement of atomic structure of a bilayer β-InSe in a unit cell. (**b**) Side view of a bilayer β-InSe. (**c**) Top view of monolayer β-InSe showing armchair and zigzag orientations. (**d**) The 2D-Brillouin zone for monolayer and few-layer β-InSe. The atoms are denoted as In (indigo color) and Se (green color) in the picture.

**Figure 2 nanomaterials-09-00082-f002:**
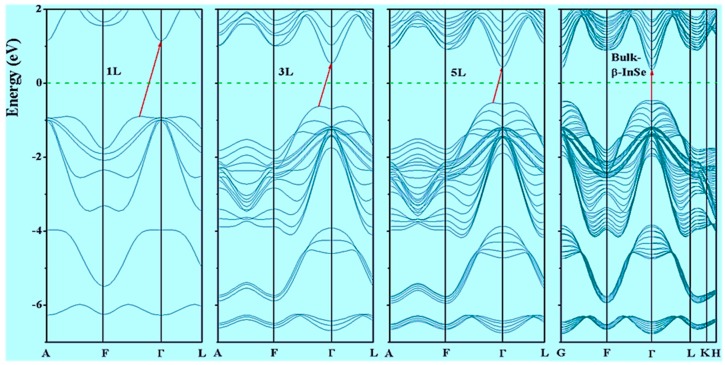
Electronic band structures of β-InSe monolayer (1 L), few-layer (3 L and 5 L) and bulk β-InSe, extracted from HSE06 functional calculations. The green dashed line is Fermi energy level set to 0.0 eV.

**Figure 3 nanomaterials-09-00082-f003:**
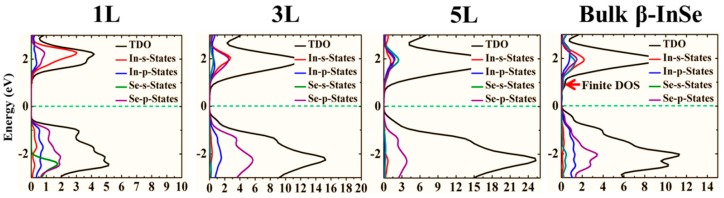
Calculated total density of states (TDOS) and partial density of states (PDOS) of β-InSe monolayer (1 L), few-layer (3 L and 5 L) and bulk β-InSe. The green horizontal dashed line represents Fermi energy level set to be 0 eV.

**Figure 4 nanomaterials-09-00082-f004:**
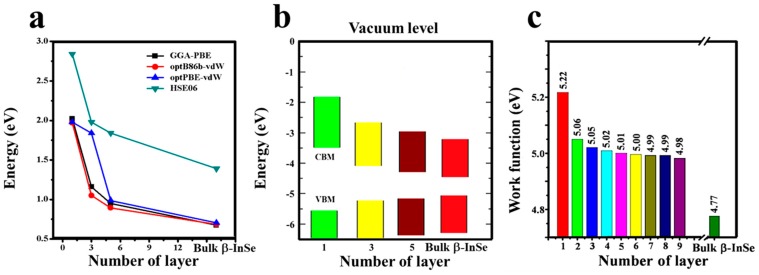
(**a**) Band gap energy of monolayer (1 L), few-layer (3 L and 5 L) and bulk β-InSe as a function of number of layers from various functionals (**b**) Schematic representation of band alignment of monolayer (1 L), few-layer β-InSe (3 L and 5 L) and bulk β-InSe, determined from HSE06 calculations. Vacuum is set as zero for reference. (**c**) Work function of few-layer β-InSe as a function of number of layers.

**Figure 5 nanomaterials-09-00082-f005:**
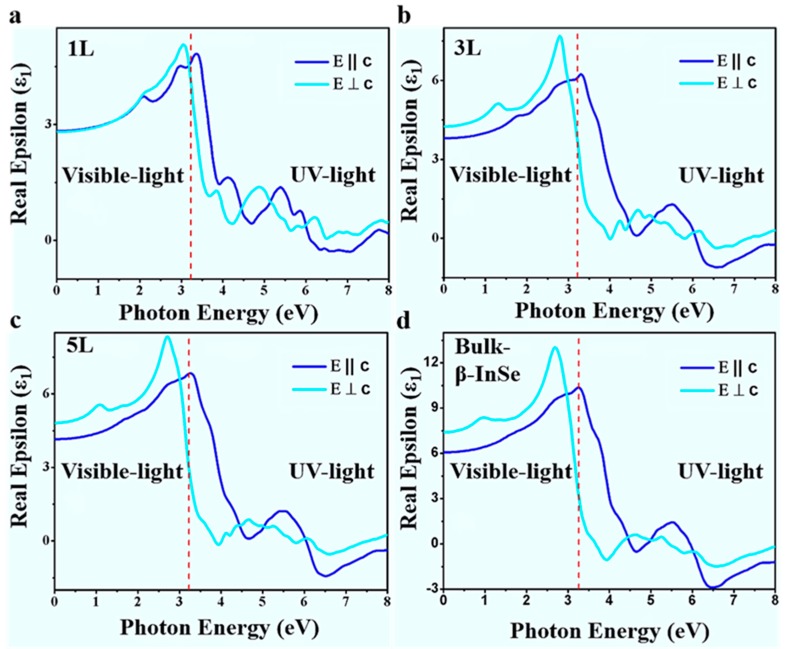
(**a**–**d**) Calculated real part of the dielectric function a long x and z directions for β-InSe monolayer (1 L), few-layer (3 L and 5 L) and bulk β-InSe.

**Figure 6 nanomaterials-09-00082-f006:**
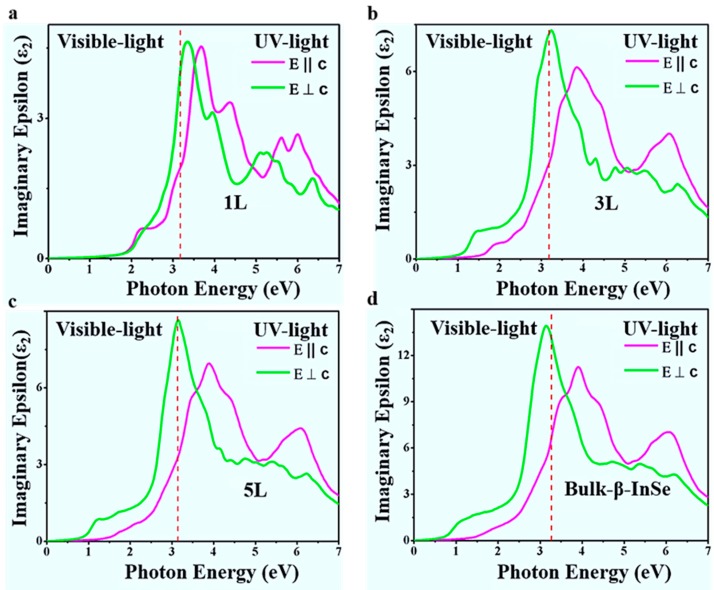
(**a**–**d**) Calculated imaginary part of the dielectric function a long x and z directions for β-InSe monolayer (1 L), few-layer (3 L and 5 L) and bulk β-InSe.

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
