# Peer review of "Two Dimensional β-InSe with Layer-Dependent Properties: Band Alignment, Work Function and Optical Properties"

_nanomaterials, 2019, doi:10.3390/nano9010082_

Round 1
Reviewer 1 Report
The paper by San et al. reports another study on InSe band structure as a function of thickness. This manuscript could be in principle taken in consideration for publication, but several issues should be addressed before acceptance.
Sanchez-Royo et al. previously reported a similar paper with a similar approach [1]. Authors should cite this work and put in evidence similarities and differences.
Figure 2 shows band-structure calculations. However, theoretical results are not compared with available angle-resolved photoemission electron spectroscopy (ARPES) experiments reported in Ref. [2, 3], where also density of states is reported.
An important milestone is the achievement of liquid-phase exfoliated indium-selenide flakes [4-6]. However, such a result is not properly discussed in this manuscript.
The recent paper on InSe-based photosensors should be cited.
A review paper on InSe should be cited [7].
The authors should report that InSe flakes are not prone to oxidation [8-12], even if Se vacancies introduces chemical reactivity toward water [8].
The first sentences of the Introduction are too trivial. Authors should focus on the necessity to replace silicon, the opportunities and pitfalls of layered semiconductors and their potential applications.
[1] Electronic structure, optical properties, and lattice dynamics in atomically thin indium selenide flakes, Nano Res. 7 (2014) 1556.
[2] Indium selenide: an insight on electronic band structure and surface excitations, Sci. Rep. 7 (2017) 3445.
[3] Thickness-dependent transition of the valence band shape from parabolic to Mexican-hat-like in the MBE grown InSe ultrathin films, Appl. Phys. Lett. 112 (2018) 191602.
[4] Solution‐Based Processing of Optoelectronically Active Indium Selenide, Adv. Mater. (2018) 1802990.
[5] Liquid‐Phase Exfoliated Indium–Selenide Flakes and Their Application in Hydrogen Evolution Reaction, Small (2018) 1800749.
[6] High‐Performance Photo‐Electrochemical Photodetector Based on Liquid‐Exfoliated Few‐Layered InSe Nanosheets with Enhanced Stability, Adv. Funct. Mater. 28 (2018) 1705237.
[7] The Advent of Indium Selenide: Synthesis, Electronic Properties, Ambient Stability and Applications, Nanomaterials 7 (2017) 372.
[8] The influence of chemical reactivity of surface defects on ambient-stable InSe-based nanodevices, Nanoscale 8 (2016) 8474.
[9] Defects and oxidation resilience in InSe, Phys. Rev. B 96 (2017) 054112.
[10] Computational mining of photocatalysts for water splitting hydrogen production: two-dimensional InSe-family monolayers, Catalysis Science & Technology 7 (2017) 2744.
[11] High-Mobility InSe Transistors: The Role of Surface Oxides, ACS Nano 11 (2017) 7362.
[12] Charge Transfer and Functionalization of Monolayer InSe by Physisorption of Small Molecules for Gas Sensing, J. Phys. Chem. C 121 (2017) 10182.
Author Response
Comment #1(Comments and suggestion for Authors): The paper by Sang et al. reports another study on InSe band structure as a function of thickness. This manuscript could be in principle taken in consideration for publication, but several issues should be addressed before acceptance.
Question 1: Sanchez-Royo et al. previously reported a similar paper with a similar approach [1]. Authors should cite this work and put in evidence similarities and differences.
Reply 1: Thank you very much for your good suggestion. The work by SanChez-Royo et al. has been cited appropriately in the revised manuscript as “Sanchez-Royo et al. previously reported a similar work [36] , and found that there was a huge increase of electronic band gap by more than 1 eV for a single layer and is in agreement with this work, despite the fact that they were investigating the band structures of γ-InSe using DFT approach as implemented in SIESTA code [37]”
Question 2: Figure 2 shows band-structure calculations. However, theoretical results are not compared with available angle-resolved photoemission electron spectroscopy (ARPES) experiments reported in Ref. [2, 3], where also density of states is reported.
Reply 2: Thank you very much for your good suggestion. Comparison of results with available experimental results has been cited as per your suggestion and the sentence has been added in the revised manuscript as “It is observed that, bulk β-InSe possess direct allowed transition and its band gap energy is comparable with the reported experimental results (1.2~1.30 eV) obtained by use of photoemission electron spectroscopy (ARPES) [50-51]”
Question 3: An important milestone is the achievement of liquid-phase exfoliated indium-selenide flakes [4-6]. However, such a result is not properly discussed in this manuscript.
Reply 3: Thank you very much for your good suggestion. The important milestone of liquid-phase exfoliated indium selenide flakes have been discussed and cited appropriately in the revised manuscript as “The greatest milestone is the achievement of liquid-phase exfoliated InSe flakes [21-23], which has provided a platform to investigate exciton physics phenomena and engineered practical devices for novel applications in optoelectronic and nanoelectronic field”
Question 4: The recent paper on InSe-based photosensors should be cited
Reply 4: Thank you very much for your suggestion. The papers on the photosensors has been discussed and cited in the revised manuscript as follows (1) “Recently, InSe was added into the family of layered transitional metal monochalcogenide. With high photoresponsivity, excellent electrical properties, and nonlinear effect, InSe has captured a lot of attention in the last few years [17-20].” (2) “The ε–InSe has ACAC layer arrangement ,and exhibits indirect band structure with a band gap value of 1.4 eV with high photoresponsivity of 34.7 mA/W in few-layer structure [35].”
Question 5: A review paper on InSe should be cited [7]
Reply 5: Thank you very much for your good suggestion. The review paper has been appropriately cited in the revised manuscript as follows “Recently, Synthesis, electronic properties, ambient stability and applications of InSe was reported [29].”
Question 5: The authors should report that InSe flakes are not prone to oxidation [8-12], even if Se vacancies introduces chemical reactivity toward water [8].
Reply 5: Thank you very much for your good suggestion. This is a very important aspect considered and mentioned in the revised manuscript as “The InSe flakes are not susceptible to oxidation [24-28] ,even if the Se vacancies induces chemical reactivity towards water [24].”
Question 6: The first sentences of the Introduction are too trivial. Authors should focus on the necessity to replace silicon, the opportunities and pitfalls of layered semiconductors and their potential applications
Reply 6: Thank you very much for your good suggestion. The insightful suggestions on the introduction of this work have been highly appreciated. The trivial description in the Introduction part has been deleted (the first sentence). The introduction part has been revised appropriately and the opportunities of the layered materials which have the layer-dependent properties like InSe but lack ambient stability were mentioned. The most iconic feature of β-InSe is stability (not prone to oxidation) and that was the opportunity. Most of the 2D materials faces the problem of oxidation and thus make it difficult to be use in the electronic and optoelectronic devices. However, with the emergence of the InSe as semiconductor material, it sizes the opportunity with a great potential to be utilized in the field of nanoelectronic and optoelectronic fields. Therefore, the revised manuscript has been tailored to capture the trend as per your guidance.
Reviewer 2 Report
This manuscript by Sang et al. aims to study the band alignment, work function and optical properties of beta(B)-InSe via theoretical DFT calculations. Overall, this is a detailed and interesting study on B-InSe which is a relatively new material in the family of 2D semiconductors. The theoretical results described by the authors can provide good guidelines to promote further experimental work on B-InSe. However, I find that there are several issues with this manuscript and the manuscript is unfit for publication in Nanomaterials in its current form. The authors are encouraged to resubmit their manuscript for consideration after appropriately addressing my concerns. Here are some of my major concerns:
1. The English language used in this manuscript is extremely poor and needs extensive revision. There are several typos, grammatical errors as well as sentence structuring errors all throughout the manuscript, which often makes it confusing for the reader. Here are some examples:
a. The first line of the Abstract can be better written as "...calculations of the layer(L)-dependent electronic band structure, work function and optical properties of B-InSe have been reported."
b. The last of the Abstract is confusing and can be written in a better way.
c. The first line of Introduction should be written as "The 2010 Nobel Prize in Physics was awarded jointly to Andre Geim..."
d. The second line in the last paragraph of the Introduction should be written in a better way.
e. The last paragraph of the Methods section is confusing. The first two sentences should be merged to form a single sentence. Currently, the second sentence "To observe changes in....periodic boundary constraints environments." doesn't make sense.
f. Page 3, Line 117: It should read: "...is very significant since it gives an insightful description of the system."
g. Page 4, Lines 151-153: It can be better written as "The band gap dependence on layer thickness offers tunability of the band gap as well as the corresponding associated..."
h. Page 5, Line 160: The first line of this paragraph is incomplete. Please rewrite properly.
There are just way too many grammatical and language errors in this manuscript which makes it extremely difficult for the reader to smoothly read through the manuscript. The authors are strongly advised to get their manuscript thoroughly proofread by an English language expert to improve its quality. In its current form, the manuscript cannot be published on the basis of poor language itself, let alone other factors.
2. The description of Figure 3 is extremely confusing. The authors state that the bottom of valence band has contribution from both s states of In and p states of Se, with little contribution from p states of In. However, looking at Figure 3, it seems that the bottom of valence band does have a significant contribution from the p states of In as indicated by the 'blue' curves which represents 'In-P-States'. The authors need to explain this discrepancy properly.
3. The quality of Figure 3 as well as the inset figures in Figure 3 needs to be improved. The inset figures need to be made larger and the font sizes of the axis labels should be increased so it is easily visible to the reader. Also, it is hard to distinguish between the red and the pink curves in Figure 3 specially when they overlap - please use a better color coding.
4. The supplementary Figure S2b is confusing. What do sub-figures (a), (b), (c) and (d) refer to? What do the inset figures represent? Please provide more details. Also, please refer to Figure S2a and Figure S2b in the main manuscript giving proper explanations.5. The authors should provide more "Keywords".
6. The authors mention about three phases of InSe, namely, the beta, gamma and the epsilon phases. However, the authors only give description of the gamma and beta phases. They should write a little about the epsilon phase to compare and contrast it to the other phases.
Author Response
Question 1: The English language used in this manuscript is extremely poor and needs extensive revision. There are several typos, grammatical errors as well as sentence structuring errors all throughout the manuscript, which often makes it confusing for the reader. Here are some examples.
Reply 1: Thank you very much for your good suggestion. All grammatical errors that you pointed has been corrected according to your suggestions and, a thorough scrutiny of the entire manuscript was carried out to ensure no typos and grammatical error. The English language in the whole manuscript has been improved.
Question 2: The description of Figure 3 is extremely confusing. The authors state that the bottom of valence band has contribution from both s states of In and p states of Se, with little contribution from p states of In. However, looking at Figure 3, it seems that the bottom of valence band does have a significant contribution from the p states of In as indicated by the blue’ curves which represents 'In-P-States'. The authors need to explain this discrepancy properly.
Reply 2: Thank you very much for your good suggestion. The wrong description on the state of the valence band has been corrected as follow: “the states at the bottom of the VBM have contribution from both p states of In and Se atoms, with little contribution from s states of In and Se atoms.”
Question 3: The quality of Figure 3 as well as the inset figures in Figure 3 needs to be improved. The inset figures need to be made larger and the font sizes of the axis labels should be increased so it is easily visible to the reader. Also, it is hard to distinguish between the red and the pink curves in Figure 3 especially when they overlap - please use a better color coding.
Reply 3: Thank you very much for your good suggestion. The quality of Figure 3 and its visibility is improved as follow. The inset was removed since the DOS curves are now visible enough and the zoomed parts are not necessary because it overcrowd the figure. Also, The color of Se-p states has been changed from pink to violet to improve the visibility on the part where there is overlap.
Figure 3. Calculated total density of states (TDOS) and partial density of states (PDOS) of β-InSe monolayer (1 L), few-layer (3 L and 5 L) and bulk β-InSe. The green horizontal dashed line represents Fermi energy level set to be 0 eV.
Question 4: The supplementary Figure S2b is confusing. What do sub-figures (a), (b), (c) and (d) refer to? What do the inset figures represent? Please provide more details. Also, please refer to Figure S2a and Figure S2b in the main manuscript giving proper explanations.
Reply 4: Thank you very much for your good suggestion. Figure S2b was for comparing individual state because of the problem of visibility. After improving the quality of Figure 3, it was found that Figure S2b serves no purpose and therefore, was removed from the revised supporting information. The description of the DOS in Figure S2 of the revised supporting information has been added as follow.
“The trend in the DOS calculation shows that 2D-DOS remained almost constant, giving insights that the electronic states of β-InSe monolayer (1 L), few-layer (2 L to 9 L) have weak dependence on the layer thickness as shown in Figure S2.”
Question 5: The authors should provide more "Keywords".
Reply 5: Thank you very much for your good suggestion. Two more Keywords have been added and the Keywords in the revised manuscript are as follow. “Layer-dependent; Indium Selenide; density functional theory; work function; optical properties”
Question 6: The authors mention about three phases of InSe, namely, the beta, gamma and the epsilon phases. However, the authors only give description of the gamma and beta phases. They should write a little about the epsilon phase to compare it to the other phases.
Reply 6: Thank you very much for your good suggestion. Indeed, mentioning the epsilon phases is very important and giving a little description of it brings a full understanding of the reason behind on the choice of β-InSe. The description of epsilon phases was added as follow. “The ε–InSe has ACAC layer arrangement ,and its Few-layer exhibit indirect band structure with a band gap value of 1.4 eV with high photoresponsivity of 34.7 mA/W [35].”
Reviewer 3 Report
The manuscript by Sang et al. reports the comprehensive data of DFT calculations on the electronic and optical properties of beta-InSe nanofilms. The Authors analyze the trends in these properties depending on the thickness (from single layer to the bulk). The subject of the study belongs to the topics of Nanomaterials. The study itself is performed by means of standard calculational scheme and on the sufficiently high technical level. Therefore, the results seem to be trustworthy in framework of accepted assumptions. In my opinion, the text contains many parts of a textbook character, which makes it too lengthy and overcrowded. Though, it is a matter of taste. I would recommend to accept the manuscript for publication after revision following the comments below.
1) In fact, the Authors have demonstrated a very weak dependence of the work function on the thickness of a few-layered films. At least, the difference between the values is below the experimentally achievable precision. The most prominent difference is found between the films and the bulk. Therefore, it would be interesting to estimate the transparency of the compound from the calculated optical properties. How many micrometers are required to absorb fully a monochromatic light along and across the c axis of InSe crystal?
2) The Authors have neglected by several electronic phenomena, which may be very important for InSe compound. First, it is not clear, why the Authors have ignored semi-core In4d10 orbitals, which are very important for the correct description of In-In bonding? Second, In is a considerably heavy element, where the spin-orbit coupling may affect the calculated band structure (occasionally, spin-orbit coupling in Se atoms may also already split the electronic levels). The Authors should make the cross-check calculation of the band structure at least for the single layer InSe, including both factors and demonstrating their strength in the modulation of electronic structure for InSe. The result could be included to the supporting information and discussed in the main text.
3) There is no reference and no comparison with the results on similar InSe systems from Magorrian et al. ( https://journals.aps.org/prb/abstract/10.1103/PhysRevB.94.245431 )
4) Mistypes found: line 28 "Ander" should be "Andre", line 117 "give" should be "gives".
5) Please, explain in caption of Fig. 3, what is the dashed line with 0.0 eV energy.
Author Response
Question 1: In fact, the Authors have demonstrated a very weak dependence of the work function on the thickness of few-layered films. At least, the difference between the values is below the experimentally achievable precision. The most prominent difference is found between the films and the bulk. Therefore, it would be interesting to estimate the transparency of the compound from the calculated optical properties. How many micrometers are required to absorb fully a monochromatic light along and across the c axis of InSe crystal?
Reply 1: Thank you very much for your good suggestion. From the results of work function, it can be concluded that layer dependence can be categorized as from monolayer (1 L) to few-layer (2 L to 9 L) to bulk, whereby the work function in few-layer are so close and decrease monotonically such that its precision is so high compared to the achievable precision in experiment. Therefore, any result within this range of few-layer attained in the experiment can be approximated to be work function of few-layer (from 2 L to 9L ) in this work. The question on transparency range is important since it is the principal for materials that defines the wavelength range which may be useful for optical device. The empirical relation lUV = anb has been used to determine the transparency range. The estimated transparency range in the E ^ c polarization direction has been established to be 0.30633 ± 0.04103 and 0.43 mm is required to absorb fully monochromatic light. In the E || c polarization direction, the transparency has been estimated to be 0.32824 ± 0.01465 and the 0.38 mm is required to absorbed fully monochromatic light.
Question 2: The Authors have neglected by several electronic phenomena, which may be very important for InSe compound. First, it is not clear, why the Authors have ignored semi-core In4d10 orbitals, which are very important for the correct description of In-In bonding? Second, In is a considerably heavy element, where the spin-orbit coupling may affect the calculated band structure (occasionally, spin-orbit coupling in Se atoms may also already split the electronic levels). The Authors should make the cross-check calculation of the band structure at least for the single layer InSe, including both factors and demonstrating their strength in the modulation of electronic structure for InSe. The result could be included to the supporting information and discussed in the main text.
Reply 2: Thank you very much for your good suggestion. Firstly, the semi-core 4d10 Indium has been considered in the calculation but due to extra-stability in these orbitals, it not more pronounced in the electronic structure of β-InSe. In the methods, correction in the description of the valence configuration to capture the d-orbitals of the two individual elements forming the β-InSe compound has been added in the revised manuscript as follow; “The valence electrons from 4d105s25p1 for In atom and 3d104s24p4 for Se atom orbitals are included”. Secondly, consideration of the spin orbital coupling (SOC) have been cross checked as your suggestion. It has been found out that the effect of SOC is very small, and other associated physical properties of β-InSe cannot be influenced by the SOC to a greater extend. The electronic band structure extracted from GGA-PBE scheme is shown in Figure S3b of the revised supporting information. In the revised manuscript, the SOC incorporation was mentioned and cited as follow: “Magorrian et al. demonstrate that inclusion of spin orbital coupling (SOC) in InSe system has little effect on energy band gap, and in this work, it was cross checked with monolayer β-InSe and established to be in agreement, as shown in Figure S3b of the supporting information, where the difference is 0.02 eV.”
Figure S3: (a) GGA-PBE band gap energies of few-layer of β-InSe as a function of number of layer, (b) electronic band structure of monolayer β-InSe extracted from GGA-PBE calculations, band dispersion lines with red is for band with SOC (Eg = 1.40 eV) and with black is for band without SOC (Eg = 1.42 eV).
Question 3: There is no reference and no comparison with the results on similar InSe systems from Magorrian et al. (https://journals.aps.org/prb/abstract/10.1103/PhysRevB.94.245431)
Reply 3: Thank you very much for your good suggestion. The comparison of the results of this work with the results found by Maggorrian et al. has been done was found to be in a good agreement. This refence has been cited as follow as follow; “The band gap energy of 1 L (2.84 eV) is comparable with the result from tight-binding model with scissor correction [52]”
Question 4: Mistypes found: line 28 "Ander" should be "Andre", line 117 "give" should be "gives".
Reply 4: Thank you very much for your good observation for a point of correction. The first sentence in the first paragraph of the introduction has been deleted because one of one reviewer claimed it is trivial and thus was deleted in the revised manuscript. In line 117, and has been corrected in the revised manuscript as follow. “The study on electronic properties of β-InSe is very significant since it gives insightful description of the system.”
Question 5: Please, explain in caption of Fig. 3, what is the dashed line with 0.0 eV energy.
Reply 5: Thank you very much for your good observation for a point of correction. The dashed line with energy 0.0 eV has been explained in the caption of Figure 3 as follow; “The green horizontal dashed line represents Fermi energy level set to be 0 eV.”
Round 2
Reviewer 2 Report
I am happy with the overall changes made by the authors after the first round of review. They have carefully addressed all the questions and concerns I had raised and have made significant improvements to their main manuscript as well as the supporting information. The figures in the main manuscript are now easier to understand. I recommend the publication of this manuscript in Nanomaterials after the authors make some minor (but important) revisions as highlighted below:
1. The language and grammar is still very poor and needs much more improvement. For example:
a. Introduction, page 2, line 32: it should be "...layered materials (2DLM) are materials with a layered structure..."
b. Introduction, page 3, line 61: it should be "Recently, synthesis, electronic...and applications of InSe were reported..."
c. Introduction, page 3, line 74: it should be "...arrangement, and exhibits indirect..."
d. Introduction, page 4, lines 90-91: it should be "...have been demonstrated to be layer thickness-dependent and this provides a..."
e. Introduction, page 4, lines 92-93: it should be "The results from this study would provide guidelines to..."
f. Page 8, lines 192-193: it should be "Despite the fact that the valence band has p orbitals from both...which is close to the Fermi level energetically."
g. Optical properties, page 11, lines 256-257: it should be "...recognition in industrial applications, particularly in the optoelectronics field, the..."
h. Optical properties, page 11, line 258: it should be "...polarization directions are the main factors..."
i. Optical properties, page 14, line 321: it should be "Other than qualitatively, optical properties can be used quantitatively."
j. Optical properties, page 14, line 315: it should be "...layer thickness tends to change..."
k. Optical properties, page 15, line 340: it should be "...the band gap increases and decreases depending on the..."
l. Optical properties, page 15, line 342-343: it should be "...change in band gap definitely changes the...that in turn leads to emission of electrons."
m. In Figure S7 caption, it should be "...index (n) along x and z..."
n. Optical properties, page 14, lines 318-319: it should be "...and this yields weak excitons."
o. Introduction section, page 2, lines 43-44: it should be "...exhibit a thickness-induced indirect-to-direct band gap transition [6-7]. The excellent..."
p. Introduction section, page 2, lines 50: it should be "...with different thicknesses have been..."
I am still unhappy with the language used in the manuscript and highly recommend that the authors get their entire manuscript as well as the supporting information thoroughly proofread and corrected by an English language expert. A manuscript with good results must also be well written otherwise it loses its charm.
2. Please correct all formatting issues in the References section. For example, in reference #9, the '2' in 'WSe2' should be in the subscript. In reference #3, 'MoS2' is written as 'MoS 2'. Please correct such errors all throughout.
3. In the Abstract, lines 21-22, the authors should perhaps write "For optical properties, the imaginary part of the dielectric function has a strong dependence on the thickness variation."
4. Recently, a very informative review paper on MoS2/2D TMDs was published in Crystals - an MDPI journal. Here is the link: https://www.mdpi.com/2073-4352/8/8/316.
For the benefit of the readers, the authors should cite this along with reference #3 on page 2, line 35 of the Introduction section where they talk about TMDs.
Author Response
Comment: I am happy with the overall changes made by the authors after the first round of review. They have carefully addressed all the questions and concerns I had raised and have made significant improvements to their main manuscript as well as the supporting information. The figures in the main manuscript are now easier to understand. I recommend the publication of this manuscript in Nanomaterials after the authors make some minor (but important) revisions as highlighted below:
Comment 1: I am still unhappy with the language used in the manuscript and highly recommend that the authors get their entire manuscript as well as the supporting information thoroughly proofread and corrected by an English language expert. A manuscript with good results must also be well written otherwise it loses its charm.
Reply 1: Thank you very much for your kind suggestion. All suggested changes have been corrected in the revised manuscript. And the manuscript has been submitted to the Language Editing System from MDPI. I hope the quality of the language would be greatly improved by the English experts.
Comment 2: Please correct all formatting issues in the References section. For example, in reference #9, the '2' in 'WSe2' should be in the subscript. In reference #3, 'MoS2' is written as 'MoS 2'. Please correct such errors all throughout.
Reply 2: Thank you very much for your good correction. In the revised manuscript, the wrong spellings have been corrected accordingly.
Comment 3: In the Abstract, lines 21-22, the authors should perhaps write "For optical properties, the imaginary part of the dielectric function has a strong dependence on the thickness variation."
Reply 3: Thank you very much for your good correction. In the revised manuscript, the correction has been added as “For optical properties, the imaginary part of the dielectric function has a strong dependence on the thickness variation.”
Comment 4: Recently, a very informative review paper on MoS2/2D TMDs was published in Crystals - an MDPI journal. Here is the link: https://www.mdpi.com/2073-4352/8/8/316.
For the benefit of the readers, the authors should cite this along with reference #3 on page 2, line 35 of the Introduction section where they talk about TMDs.
Reply 4: Thank you very much for your good suggestion. In the revised manuscript, the reference has be cited as “Besides graphene, hexagonal boron nitride (h-BN) [2], transitional metal dichalcogenides (TMDs) [3-4],”
Reviewer 3 Report
The Authors have considered my comments seriously. The text of manuscript was revised properly. I can only regret that the Authors did not include the discussion of transparency range for other readers. Anyway, I recommend to accept the manuscript of this detailed study of the InSe films for publication in its current form.
Author Response
The Authors have considered my comments seriously. The text of manuscript was revised properly. I can only regret that the Authors did not include the discussion of transparency range for other readers. Anyway, I recommend to accept the manuscript of this detailed study of the InSe films for publication in its current form
Reply:Thank you very much for your recommendation. The corresponding discussion about the transparency range has been added to the revised manuscript as following.
“The estimated transparency range in the E ^ c polarization direction has been established to be 0.30633 ± 0.04103 and 0.43 mm thickness is required to absorb fully monochromatic light. In the E || c polarization direction, the transparency has been estimated to be 0.32824 ± 0.01465 and the 0.38 mm thickness is required to absorbed fully monochromatic light.”